# Adaptive Hyperparameter Selection for Differentially Private Gradient Descent

**Dominik Fay**                                                    *dominikf@kth.se*
*Division of Decision and Control Systems*
*KTH Royal Institute of Technology*

*Elekta*

**Sindri Magnússon**                                    *sindri.magnusson@dsv.su.se*
*Department of Computer and Systems Sciences*
*Stockholm University*

**Jens Sjölund**                                              *jens.sjolund@it.uu.se*
*Department of Information Technology*
*Uppsala University*

**Mikael Johansson**                                            *mikaelj@kth.se*
*Division of Decision and Control Systems*
*KTH Royal Institute of Technology*

**Reviewed on OpenReview:** *https://openreview.net/forum?id=LLKI5Lq2YN*

## Abstract

We present an adaptive mechanism for hyperparameter selection in differentially private optimization that addresses the inherent trade-off between utility and privacy. The mechanism eliminates the often unstructured and time-consuming manual effort of selecting hyperparameters and avoids the additional privacy costs that hyperparameter selection otherwise incurs on top of that of the actual algorithm.

We instantiate our mechanism for noisy gradient descent on non-convex, convex and strongly convex loss functions, respectively, to derive schedules for the noise variance and step size. These schedules account for the properties of the loss function and adapt to convergence metrics such as the gradient norm. When using these schedules, we show that noisy gradient descent converges at essentially the same rate as its noise-free counterpart. Numerical experiments show that the schedules consistently perform well across a range of datasets without manual tuning.

## 1 Introduction

In tandem with the successes of machine learning, driven in particular by ever larger and more data-hungry neural networks, there is mounting concern over privacy among both policymakers and the general public. Researchers have noticed, and the last few years have witnessed intense efforts at reconciling the competing demands of privacy and utility. A major line of work has focused on modifying the optimization procedure to obtain guarantees on differential privacy. These all face the question of *how to distribute the privacy budget to achieve maximum utility?* Typically, addressing this question boils down to the selection of hyperparameters that control the privacy-utility tradeoff. But ultimately finding reasonable values for these has, so far, been largely left as an exercise for the reader. In addition to being time-consuming, manual hyperparameter tuning also incurs an extra (sometimes neglected) privacy cost on top of the actual algorithm.

Differential privacy (DP) and empirical risk minimization (ERM) are two key concepts in the field of privacy-preserving machine learning. The prototypical algorithm for DP-ERM is Noisy Stochastic Gradient Descent (Song et al., 2013; Bassily et al., 2014), variants of which have been successfully applied in various domains such as medical imaging (Kaissis et al., 2021; Ziller et al., 2021; Adnan et al., 2022) and large language models McMahan et al. (2018); Basu et al. (2021). The standard result for excess empirical risk in $d$ dimensions and a sample size of $N$ is that it achieves $O(\sqrt{d}/(N\epsilon))$ for convex losses (Bassily et al., 2014), and $O(d/(\mu N^2 \epsilon^2))$ for Lipschitz-smooth $\mu$-strongly convex losses (Kifer et al., 2012) under $(\epsilon, \delta)$-DP. These bounds are worst-case optimal, i.e., they match known lower bounds (Bassily et al., 2014). Although they can be achieved with a uniform privacy budget allocation, a number of recent works have provided empirical evidence that adaptive schedules can improve performance on more typical machine learning problems, such as generalized linear problems and deep learning (Lee & Kifer, 2018; Song et al., 2020; Zhang et al., 2021; Mohapatra et al., 2022).

The main hyperparameters for Noisy SGD are the step size and the noise scale, i.e., the amount of random noise added to each gradient update. Common approaches for selecting hyperparameters for differentially private algorithms include differentially private model selection from private candidates via grid search (Yu et al., 2019) or search with random stopping (Liu & Talwar, 2019; Papernot & Steinke, 2022; Koskela & Kulkarni, 2023) as well as Bayesian optimization (Avent et al., 2020). However, these approaches are generally time-consuming and incur an additional privacy cost. In this work, we propose a simple strategy for hyperparameter selection that avoids both the additional computational cost and the privacy cost. In summary, our contributions are as follows:

1. We propose a conceptual framework for tuning time-varying hyperparameters by optimizing, at each step, the *privacy-utility ratio* (PUR).

2. We derive schedules for Lipschitz-smooth losses, showing that the optimal noise variance is proportional to the squared gradient norm. We specialize the schedules to non-convex, convex and strongly convex cases. In contrast to existing works on noisy gradient methods, the proposed schedules attain the same convergence speed as their noise-free counterparts.

3. To obtain rigorous privacy guarantees, we upper bound the gradient norm to derive data-independent versions of the above schedules while retaining the same convergence rate as the data-dependent ones.

4. Experiments across multiple datasets show that our schedules are at least as good as using an optimally tuned constant noise variance, even when the privacy cost of hyperparameter tuning is ignored.

The remainder of the paper is organized as follows: In Section 2 we provide some necessary background on convex optimization and differential privacy. In Section 3 we introduce our framework for adaptive hyperparameter selection, and present our theoretical results. We show a summary of our theoretical results in Tables 1 and 2. We complement this with experimental results in Section 4, and discuss our results in Section 5, together with suggestions for future work. We conclude with a discussion of related work in Section 6. Additional numerical results and proofs are provided in Appendices A and B, respectively.

## 2   Background

We begin by defining core concepts from convex analysis and then summarize main results from differential privacy that we later use in our analysis.

**Convex optimization**   In convex optimization we typically consider functions that have one or more of the following three properties:

**Definition 1** (*L*-Lipschitz continuity). *A function $f : \mathcal{C} \to \mathbb{R}$ is L-Lipschitz continuous if*

$$|f(y) - f(x)| \le L\|y - x\| \quad \text{for all } x, y \in \mathcal{C}.$$

**Definition 2** ($\mu$-strong convexity). *A differentiable function $f : \mathcal{C} \to \mathbb{R}$ is $\mu$-strongly convex if*

$$f(y) \geq f(x) + \langle \nabla f(x), y - x \rangle + \frac{\mu}{2} \|y - x\|^2 \quad \text{for all } x, y \in \mathcal{C}.$$

**Definition 3** ($M$-smoothness). *A differentiable function $f : \mathcal{C} \to \mathbb{R}$ is $M$-smooth if*

$$f(y) \leq f(x) + \langle \nabla f(x), y - x \rangle + \frac{M}{2} \|y - x\|^2 \quad \text{for all } x, y \in \mathcal{C}.$$

Note that if a function $f$ is differentiable and $L$-Lipschitz, then its gradient norm is bounded by $L$. If $f$ is additionally convex then its gradient is Lipschitz-continuous. Likewise, if $f$ is convex and $M$-smooth then it is also $L$-Lipschitz (Boyd & Vandenberghe, 2014). However, in either case, the best known bound on $M$ (or $L$) for any particular function $f$ may be stronger than the bound implied by $L$ (or $M$, respectively).

**Differential privacy**  The form of privacy we ultimately want to achieve is $(\epsilon, \delta)$-differential privacy, which is defined formally as follows.

**Definition 4** ($(\epsilon, \delta)$-differential privacy (Dwork et al., 2006)). *Let $\sim_X$ be a symmetric relation on a set $X$. A randomized function $\mathcal{M} : X \to Y$ is $(\epsilon, \delta)$-differentially private if for all $x \sim_X x'$ and all measurable $S \subseteq Y$,*

$$\Pr[\mathcal{M}(x) \in S] \leq e^\epsilon \Pr[\mathcal{M}(x') \in S] + \delta.$$

Most differential privacy mechanisms are based on bounding the influence of individual data points on the output of a function, which is captured by the notion of sensitivity.

**Definition 5** (Sensitivity). *Let $\sim_X$ be a symmetric relation on a set $X$. The sensitivity of a function $f : X \to Y$ with respect to $\sim_X$ is defined as*

$$\Delta = \sup_{x \sim_X x'} \|f(x) - f(x')\|.$$

In this work, we take the $\sim_X$ to be the replacement relation, i.e. $x \sim_X x'$ if $x'$ is obtained from $x$ by replacing one data entry with another. Of particular interest to ERM is the arithmetic mean $f(x_1, \ldots, x_N) = 1/N \sum_n x_n$ defined over a bounded convex set $\mathcal{C}$ which has sensitivity $\Delta = D/N$ where $D = \max_{y,z \in \mathcal{C}} \|y - z\|$ is the diameter of $\mathcal{C}$.

In the context of differential privacy, a random perturbation of a deterministic function $f(x)$ is referred to as a mechanism $\mathcal{M}(x)$. In particular, we focus on the Gaussian mechanism $\mathcal{M}(x; \sigma) = f(x) + \zeta$ where independent and identically distributed Gaussian noise $\zeta \sim \mathcal{N}(0, \sigma^2 \boldsymbol{I})$ is added to the output of a deterministic function $f$.

Apart from $(\epsilon, \delta)$-differential privacy, several variants of differential privacy have emerged that better cater to the characteristics of more restricted classes of noise distributions. In particular, the properties of the Gaussian mechanism are well-described by zero-concentrated differential privacy.

**Definition 6** (Zero-concentrated differential privacy (Bun & Steinke, 2016)). *Let $\sim_X$ be a symmetric relation on a set $X$. A randomized function $\mathcal{M} : X \to Y$ is $\rho$-zCDP if for all $x \sim_X x'$,*

$$D_\alpha(\mathcal{M}(x) \,\|\, \mathcal{M}(x')) \leq \alpha \rho \quad \text{for all } \alpha > 1,$$

*where $D_\alpha$ is the Rényi divergence of order $\alpha$.*

Specifically, the Gaussian mechanism with variance $\sigma^2$ satisfies $\rho$-zCDP for $\rho = \Delta/(2\sigma^2)$ where $\Delta$ is the sensitivity of $f$. The Gaussian mechanism also satisfies $(\epsilon, \delta)$-DP for $\epsilon > \sqrt{2 \log(1.25/\delta)} \Delta/\sigma$ and $\epsilon < 1$.

A convenient property of zCDP is that composition is linear, in other words, an adaptive sequence of mechanisms $(\mathcal{M}_i)_{i=1}^k$ jointly satisfies $\rho$-zCDP if each $\mathcal{M}_i$ satisfies $\rho_i$-zCDP and $\rho = \sum_i \rho_i$. A reference for the above claims relating to $(\epsilon, \delta)$-DP and $\rho$-zCDP can be found in e.g. Dwork & Roth (2014) and Bun & Steinke (2016), respectively. A comparison between the various notions of differential privacy can be found in Desfontaines & Pejó (2020).

| | Excess loss | Iterations | Local gradient evaluations |
|---|---|---|---|
| Bassily et al. (2014) | $\sqrt[2]{\frac{d\log^3 N}{\epsilon^2 N^2}}$ | $N^2$ | $N^2$ |
| Wang et al. (2017) | $\sqrt[2]{\frac{d}{\epsilon^2 N^2}}$ | $\log\frac{N\epsilon}{\sqrt{d}}$ | $\frac{N\epsilon}{\sqrt{d}} + N\log\frac{N\epsilon}{d}$ |
| Ours | $\sqrt[3]{\frac{d}{\epsilon^2 N^2}},\ \sqrt[3]{\frac{d}{\rho N^2}}$ | $\sqrt[3]{\frac{\epsilon N^2}{d}}$ | $N^{5/3}\sqrt[3]{\frac{\epsilon}{d}}$ |

Table 1: Comparison of DP-ERM algorithms for convex, Lipschitz, Lipschitz-smooth loss functions. The first column refers to the excess empirical risk. The second and third column refer to the number of iterations/evaluations needed to achieve the loss listed in the first column. All entries are upper bounds and should be read as $\mathcal{O}(\cdot)$ where the Lipschitz constant $L$, Lipschitz-smoothness constant $M$ and the second privacy parameter $\delta$ are treated as constants. When $\rho$ is given, it refers to a guarantee in terms of $\rho$-zCDP instead of $(\epsilon, \delta)$-DP. The dependence on $\epsilon$ is stated for the "high-privacy" regime ($\epsilon \to 0$). For the "low-privacy" regime ($\epsilon \to \infty$), replace $\epsilon^2$ with $\epsilon$.

| | Excess loss | Iterations | Local gradient evaluations |
|---|---|---|---|
| Bassily et al. (2014) | $\frac{d\log^2 N}{N^2\epsilon^2}$ | $N^2$ | $N^2$ |
| Wang et al. (2017) | $\frac{d\log N}{N^2\epsilon^2}$ | $\log\frac{N^2\epsilon^2}{d}$ | $N\log\frac{N\epsilon}{d}$ |
| Hong et al. (2022) | $\frac{d}{N^2\rho}$ | $\log\frac{N^2\rho}{d}$ | $N\log\frac{N^2\rho}{d}$ |
| Ours | $\frac{d}{N^2\epsilon^2},\ \frac{d}{N^2\rho}$ | $\log(\epsilon N^2)$ | $N\log(\epsilon N^2)$ |
| Lower bound | $\frac{d}{N^2\epsilon^2}$ | | |

Table 2: Comparison of DP-ERM algorithms for strongly convex, Lipschitz, Lipschitz-smooth loss functions. The first column refers to the excess empirical risk. The second and third column refer to the number of iterations/evaluations needed to achieve the loss listed in the first column. All entries except for the last row are upper bounds and should be read as $\mathcal{O}(\cdot)$ where the Lipschitz constant $L$, Lipschitz-smoothness constant $M$, strong convexity constant $\mu$ the second privacy parameter $\delta$ are treated as constants. When $\rho$ is given, it refers to a guarantee in terms of $\rho$-zCDP instead of $(\epsilon, \delta)$-DP. The dependence on $\epsilon$ is stated for the "high-privacy" regime ($\epsilon \to 0$). For the "low-privacy" regime ($\epsilon \to \infty$), replace $\epsilon^2$ with $\epsilon$.

## 3 Adapting Hyperparameters to the Privacy-Utility Ratio

We consider the problem of differentially private empirical risk minimization (DP-ERM). That is, we want to minimize the empirical risk

$$F(\boldsymbol{\theta}) = \frac{1}{N}\sum_n f(\boldsymbol{\theta}; \boldsymbol{x}_n) \tag{1}$$

over a parameter vector $\boldsymbol{\theta} \in \mathbb{R}^d$ for a dataset $\boldsymbol{x}_1, \ldots \boldsymbol{x}_N \in \mathcal{X}$, under the constraint that $\boldsymbol{\theta}$ preserve $(\epsilon, \delta)$-differential privacy. To this end, we revisit the differentially private gradient descent (DP-GD) algorithm, which consists of noisy gradient steps

$$\theta_{t+1} = \theta_t - \eta_t\left(\nabla F(\theta_t) + \zeta_t\right), \quad \zeta_t \sim \mathcal{N}(0, \sigma_t^2 \boldsymbol{I}) \tag{2}$$

with time-varying step sizes $\eta_t$ and noise variances $\sigma_t^2$.

### 3.1 Main Idea

The main idea is to select the step size $\eta_t$ and noise standard deviation $\sigma_t$ that, at each step, minimize the privacy loss per unit of utility improvement. We call this the privacy-utility ratio (PUR) and define it as

$$\text{PUR}(\sigma_t, \eta_t) = \frac{P(\sigma_t)}{U(\sigma_t, \eta_t)}, \tag{3}$$

for suitably chosen functions $U(\sigma_t, \eta_t)$ and $P(\sigma_t)$ corresponding, respectively, to utility improvement and privacy cost. The utility function can incorporate convergence information such as the gradient norm or objective value. Thereby, minimizing the PUR allows us to adapt the privacy budget to the optimization progress. For instance, we might expect that later stages of the optimization require higher precision since, typically, the gradient norm tends to zero as we approach the optimum.

We measure the utility improvement $U(\sigma_t, \eta_t)$ via a descent lemma that bounds the expected loss improvement in the next iteration, which can be derived from analytical properties of the loss function $F$. Although, the associated privacy cost $P(\sigma_t)$ is independent of the step size $\eta_t$, its exact form depends on the variant of differential privacy we choose to apply. Again, we ultimately want to ascertain $(\epsilon, \delta)$-differential privacy, which permits a simple expression for the privacy cost, see Section 3.2.

Based on the above choices of utility and privacy, we derive step-wise optimal schedules for selecting the hyperparameters ($\eta_t$ and $\sigma_t$) and analyze their convergence rates. In Section 3.3, we first consider the setting where the utility improvement depends directly on convergence information such as the gradient norm. This is, however, an idealized setting since the gradient norm itself is data-dependent and hence sensitive information. In order to overcome this limitation, in Section 3.4, we replace this dependence with a bound to arrive at a data-independent schedule. Curiously, our analysis shows that the data-independent schedule attains essentially the same convergence rate as the data-dependent one.

### 3.2 Privacy cost

Our approach to deriving a privacy cost function is based on the $(\epsilon, \delta)$-DP privacy cost of the Gaussian mechanism under a Lipschitz assumption on the loss function, which is a common approach in the literature (Song et al., 2013; Bassily et al., 2014). If the example-level loss $f(\,\cdot\,; \boldsymbol{x}_n)$ is $L$-Lipschitz for all $\boldsymbol{x}_n$, then it follows from Equation 1 that the full gradient $\nabla F$ of the empirical risk has sensitivity $2L/N$. Therefore, by the classical analysis of the Gaussian mechanism, $\theta_{t+1}$ computed via Equation 2 from $\theta_t$ preserves $(\epsilon, \delta)$-DP for any

$$\sigma_t > \sqrt{2 \log(1.25 \delta^{-1})} \, \frac{2L}{N\epsilon}, \quad \epsilon < 1.$$

Note that the constraint $\epsilon < 1$ only needs to be satisfied for individual iterations. As long as a reasonable ("single-digit") total privacy budget is imposed, it is unlikely that we violate this constraint, given that we can expect to perform a large number of iterations. For the sake of tractability, we choose to drop the constraint and define our privacy cost function as

$$P(\sigma_t) = \frac{c}{\sigma_t} \quad \text{with } c = \sqrt{2 \log(1.25 \delta^{-1})} \frac{2L}{N}. \tag{4}$$

We emphasize that the constraint is enforced in our subsequent privacy analysis, it is only dropped while we develop a suitable heuristic.

### 3.3 Data-dependent selection

We use the assumption that $F$ is $M$-smooth to formulate a descent lemma for estimating the expected improvement in the objective function for a given step-size and noise variance. Specifically, for a single update step, we have the following result:

**Lemma 1.** *Let $F$ be $M$-smooth. If $\theta_{t+1}$ is computed via Equation 2, then*

$$\mathbb{E}\left[F(\theta_t) - F(\theta_{t+1}) \mid \theta_t, \sigma_t\right] \geq \left(\eta_t - \frac{M}{2}\eta_t^2\right) \|\nabla F(\theta_t)\|^2 - \frac{M}{2}\eta_t^2 d\sigma_t^2. \tag{5}$$

We use the lower bound on the expected improvement from Lemma 1 as our utility function,

$$U(\sigma_t, \eta_t; \boldsymbol{\theta}) = \left(\eta_t - \frac{M}{2}\eta_t^2\right)\|\nabla F(\boldsymbol{\theta})\|^2 - \frac{M}{2}\eta_t^2 d\sigma_t^2.$$

Note that the need to evaluate the gradient norm $\|\nabla F(\boldsymbol{\theta})\|$ makes this function data dependent.

Equipped with this utility function and the privacy cost from Equation 4 we can find the hyperparameters that minimize the privacy-utility ratio. The result is captured by the following proposition:

**Proposition 1** (Data-dependent schedule). *The privacy-utility ratio* $\mathrm{PUR}(\sigma_t, \eta_t)$ *is minimized by*

$$\sigma_t = \frac{\|\nabla F(\theta_t)\|}{\sqrt{d}} \quad and \quad \eta_t = \frac{1}{2M}. \tag{6}$$

There are multiple observations worth highlighting about this schedule:

- First, it is remarkably simple – the step size is constant and the noise standard deviation is directly proportional to the gradient norm. The reason for the former is that the optimal step size for an arbitrary $\sigma_t$ depends on the "signal-to-noise ratio" $\|\nabla F(\theta_t)\|^2/\sigma_t^2$ (see Equation 12). When the noise standard deviation is proportional to the gradient norm, the signal-to-noise ratio is constant and therefore the optimal step size is constant.

- Second, most of the prior work on differentially private gradient-based optimization considers a decaying step size and a constant noise variance. In contrast, Proposition 1 suggests that the roles should be reversed – the step size should be constant and the noise variance should be decaying.

- Third, the schedule is independent of the privacy parameters $\epsilon$ and $\delta$, which means that the chosen privacy budget determines the time horizon $T$. Again, this contrasts with prior work which fixes the time horizon $T$ and scales the noise variance to meet the privacy budget.

- Finally, the schedule is data dependent, because the gradient norm $\|\nabla F(\theta_t)\|$ is required to compute the noise standard deviation. This is not particularly surprising given that we allowed the expected utility improvement $U$ to depend on $\theta_t$, but it has an important practical implication: The schedule itself exhibits a privacy leakage that must be accounted for. This is the subject of Section 3.4.

Before moving on to the data-independent schedule, we first state the convergence rate of the algorithm when using the above schedule, assuming oracle access to the gradient norm $\|\nabla F(\theta_t)\|$.

**Proposition 2** (Data-dependent convergence rate). *Let $F$ be $M$-smooth and $\theta_{t+1}$ be computed recursively via Equation 2 with $\eta_t = 1/(2M)$ and $\sigma_t = \|\nabla F(\theta_t)\|/\sqrt{d}$, then*

*(a) for (possibly) non-convex $F$,*

$$\mathbb{E}\left[\frac{1}{T}\sum_{t=1}^{T}\|\nabla F(\theta_t)\|^2\right] \le \frac{4M\left(F(\theta_0) - F^*\right)}{T}, \tag{7}$$

*(b) if $F$ is convex and the iterates satisfy $\|\theta_t - \theta^*\| \le R$,*

$$\mathbb{E}\left[F(\theta_T) - F^*\right] \le \frac{4MR^2}{T}, \tag{8}$$

*(c) if $F$ is $\mu$-strongly convex,*

$$\mathbb{E}\left[F(\theta_T) - F^*\right] \le \left(1 - \frac{\mu}{2M}\right)^T \left(F(\theta_0) - F^*\right), \tag{9}$$

*where $F^*$ is the minimal empirical risk.*

The convergence rates are remarkably close to those for the non-private gradient descent algorithm: for convex losses, the convergence rate is $\mathcal{O}(1/t)$ in both cases. For strongly convex losses, it is $\mathcal{O}(r^t)$ where $r = 1 - \mu/M$ in the non-private case and $r = 1 - \mu/(2M)$ in the private case, meaning that the private version only needs approximately twice as many iterations to reach the same accuracy. This is because $\log(1 - \mu/M) \approx 2\log(1 - \mu/(2M))$, unless $M/\mu$ is very small.

### 3.4 Data-independent selection

In this section, we derive a data-independent version of the PUR-optimal schedule and analyze its convergence. In summary, the main results are that the data-independent schedule (a) converges at essentially the same rate as the data-dependent schedule in terms of iterations, and (b) has similar privacy-utility convergence as recent work on strongly convex losses (Hong et al., 2022), while additionally permitting an upper bound on non-convex and (non-strongly) convex losses.

We begin by deriving the data-independent schedule. Recall from Proposition 1 that the PUR-optimal schedule $\sigma_t$ at time $t$ is proportional to the gradient norm $\|\nabla F(\theta_t)\|$. Proposition 2 shows that this schedule exhibits essentially the same convergence rate as non-private GD. While this bound is stated in terms of excess risk, similar results are known for the gradient norm in non-private GD: the gradient norm converges at a rate of $\mathcal{O}(1/t)$ for convex losses and $\mathcal{O}(r^t)$ for strongly convex losses, where $r = 1 - \mu/M$. The idea for a data-independent schedule is then to use these upper bounds as a proxy for the gradient norm itself. This leads to the following result.

**Proposition 3** (Data-independent convergence rate). *Let $F$ be $M$-smooth and $\theta_{t+1}$ computed via Equation 2 with $\eta_t = 1/(2M)$, then*

(a) *if $\sigma_t = \frac{4M\sqrt{F(\theta_0) - F^*}}{\sqrt{dt}}$,*

$$\mathbb{E}\left[\frac{1}{T}\sum_{t=0}^{T}\|\nabla F(\theta_t)\|^2\right] \leq \frac{8(\pi^2 M/3 + 1)(F(\theta_0) - F^*)}{3T},$$

(b) *if $F$ is convex, $\sigma_t = \frac{4MR}{\sqrt{dt}}$ and the iterates satisfy $\|\theta_t - \theta^*\| \leq R$,*

$$\mathbb{E}\left[F(\theta_T) - F^*\right] \leq \frac{16MR^2}{3T},$$

(c) *if $F$ is $\mu$-strongly convex and $\sigma_t = \sqrt{2\mu\left(F(\theta_0) - F^*\right)\left(1 - \mu/(2M)\right)^t/d}$,*

$$\mathbb{E}\left[F(\theta_T) - F^*\right] \leq \left(1 - \frac{\mu}{2M}\right)^T (F(\theta_0) - F^*), \tag{10}$$

*where $F^*$ is the minimal empirical risk.*

Remarkably, the data-independent schedule achieves the same convergence rate as the data-dependent schedule, up to a small multiplicative factor.

Having removed the data dependence, we are now able to analyze the privacy loss of the algorithm. The analysis follows standard arguments: Each iteration has constant sensitivity, and the noise variance from Proposition 3 is such that releasing the noisy gradient satisfies $\mathcal{O}(t^2)$-zCDP and $\mathcal{O}((1/r)^t)$-zCDP, respectively in the convex and strongly convex cases. Accumulating the privacy loss across $T$ iterations, and combining the result with Proposition 3, yields the following proposition.

**Proposition 4** (Privacy-utility convergence). *Let $F$ be $M$-smooth and $f$ be $L$-Lipschitz. If $\theta_{t+1}$ is computed via Equation 2 with $\eta_t = 1/(2M)$, then the iterates $\theta_1, \ldots, \theta_T$ jointly satisfy $(\epsilon, \delta)$-DP. In particular,*

(a) *if $\sigma_t = \frac{4M\sqrt{F(\theta_0) - F^*}}{\sqrt{dt}}$, then, after $T = \sqrt[3]{8\rho N^2 M^2 (F(\theta_0) - F^*)/(L^2 d)}$ iterations,*

$$\mathbb{E}\left[\frac{1}{T}\sum_{t=0}^{T}\|\nabla F(\theta_t)\|^2\right] \leq \frac{8}{3}\left(\frac{\pi^2 M}{3} + 1\right)\sqrt[3]{\frac{(F(\theta_0) - F^*)^2 L^2 d}{8\rho N^2 M^2}}.$$

(b) *if $F$ is convex, $\sigma_t = \frac{4MR}{\sqrt{dt}}$ and the iterates satisfy $\|\theta_t - \theta^*\| \leq R$, then, after $T = \sqrt[3]{8\rho N^2 M^2 R^2/(L^2 d)}$ iterations,*

$$\mathbb{E}\left[F(\theta_T) - F^*\right] \leq \frac{8}{3}\sqrt[3]{\frac{L^2 MRd}{\rho N^2}}.$$

(c) *if $F$ is $\mu$-strongly convex and $\sigma_t = \sqrt{2\mu\left(F(\theta_0) - F^*\right)\left(1 - \mu/(2M)\right)^t/d}$, then, after $T = \left(\log \frac{\rho\mu(F(\theta_0)-F^*)N^2}{2dL^2}\right)/\left(\log \frac{2\kappa}{2\kappa-1}\right)$ iterations,*

$$\mathbb{E}\left[F(\theta_T) - F^*\right] \leq \frac{L^2 d}{N^2 \mu \rho}, \tag{11}$$

*where $\kappa = M/\mu$.*

*Here, $F^*$ is the minimal empirical risk and $\epsilon = \rho + 2\sqrt{\rho \log \delta^{-1}}$.*

The upper bound on strongly convex losses is optimal, in the sense that it is on the same order as known lower bounds (Bassily et al., 2014). In contrast to previous work that also achieved this (Hong et al., 2022), we additionally have an upper bound on convex and non-convex losses. Note that the convex upper bound scales with $\sqrt[3]{\frac{d}{\rho N^2}}$. This is because the privacy loss grows as $\mathcal{O}(T^3)$ while the excess loss, even in the non-private case, only decreases as $\mathcal{O}(1/T)$.

We provide a comparison between Proposition 4 and results from previous work in Tables 1 and 2 in terms of simplified order bounds.

## 4 Experiments

We evaluate the performance of the proposed hyperparameter schedules on synthetic and real-world datasets, on convex and strongly convex loss functions. The primary purpose of our experiments is to verify whether the proposed automatic hyperparameter selection can consistently outperform hyperparameters found via exhaustive search.

**Loss function** We consider regularized logistic regression

$$f(\boldsymbol{\theta}; \boldsymbol{z}_n, y_n) = \log\left(1 + \exp(-y_n \boldsymbol{z}_n^\top \boldsymbol{\theta})\right) + \frac{\lambda}{2}\|\boldsymbol{\theta}\|^2$$

with feature vectors $\boldsymbol{z}_n \in \{\boldsymbol{z} \in \mathbb{R}^d \mid \|\boldsymbol{z}\| \leq Z\}$ and labels $y_n \in \{-1, 1\}$ and regularization parameter $\lambda \geq 0$. The corresponding empirical risk $F$ is convex, $L$-Lipschitz and $M$-smooth with $L = \lambda R + Z$ and $M = \lambda + Z^2/4$, where $R$ is an upper bound on $\|\theta_t - \theta^*\|$. If $\lambda > 0$ then $F$ is also $\lambda$-strongly convex.

**Baselines** We compare the privacy-utility performance of our adaptive, data-independent schedules (cf. Proposition 3) to that of the constant schedule $\sigma_t = \sigma$ for a wide range of values of $\sigma \in \{0.001, 0.01, 0.1, 1.0\}$. Furthermore, we include a non-gradient-descent baseline, namely output perturbation, which first solves the optimization problem non-privately via an arbitrary solver and then adds noise to the solution. In particular, we use the black-box output perturbation algorithm for convex, Lipschitz ERM objectives from Lowy & Razaviyayn (2021, Algorithm 6). Finally, we also show the hypothetical privacy-utility performance of the data-dependent schedule (Equation 6), assuming oracle access to the gradient norm.

Regarding the data-independent schedule, note that the regularization parameter $\lambda$ determines which schedule we apply: The schedule for strongly convex losses is used when $\lambda > 0$, and the schedule for convex losses for $\lambda = 0$. We use the same step size $\eta_t = 1/(2M)$ in all runs. The privacy cost is computed in the same way for all gradient perturbation methods: The per-iteration costs are aggregated via zCDP composition, and then converted to $(\epsilon, 1/N)$-differential privacy. For output perturbation, we use the privacy analysis of Lowy & Razaviyayn (2021).

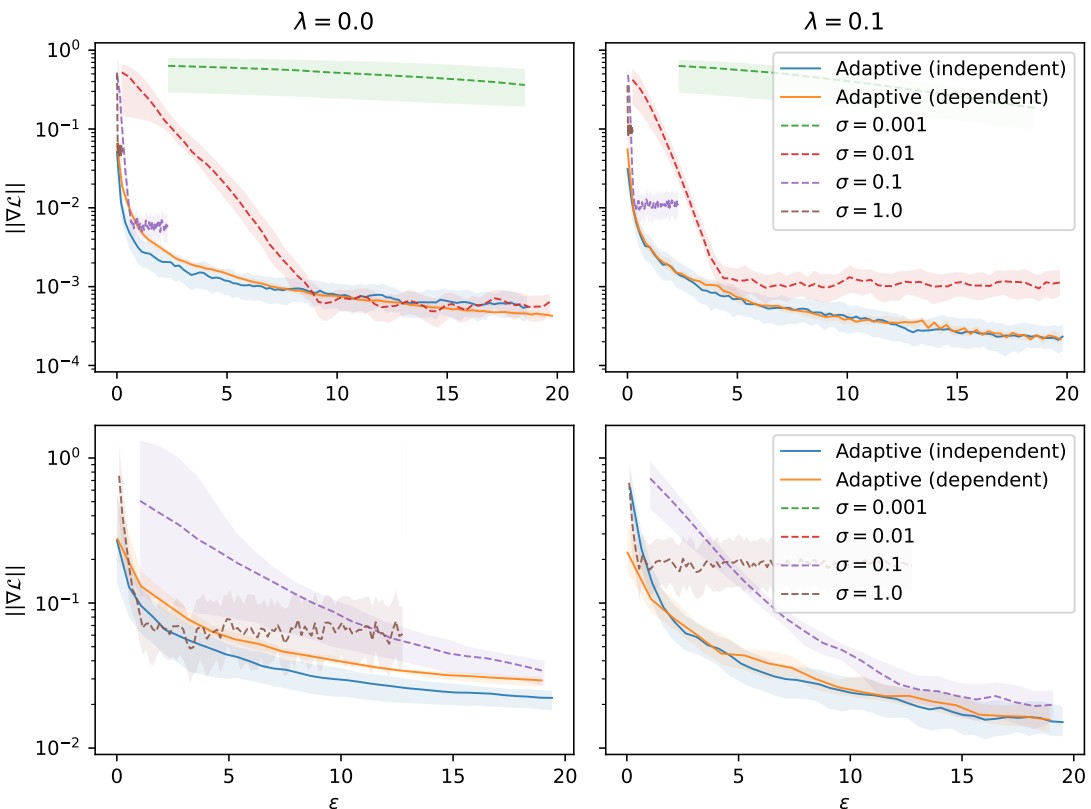

Figure 1: Convergence of the gradient plotted against privacy expenditure for various noise schedules. The lines shown are the median of 120 repetitions. The shaded area is the inter-quartile range. Some schedules exceed the maximum number of iterations before reaching $\epsilon = 20$. The data-dependent schedule assumes oracle access to the gradient norm. Top: Synthetic dataset. Bottom: Iris dataset. Left column: Convex objective ($\lambda = 0$). Right column: Strongly convex objective ($\lambda = 0.1$).

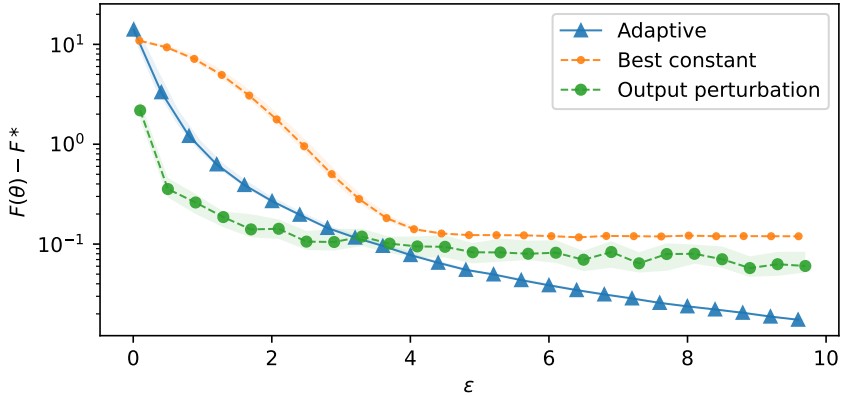

Figure 2: Excess loss for the CIFAR-10 dataset with ScatterNet features. *Best constant* refers to the constant schedule with best noise variance in hindsight. The shaded area is the inter-quartile range.

**Datasets**   We repeat our experiments on six different datasets. All datasets are preprocessed such that the feature vectors $\mathbf{z}_n$ have zero mean and unit variance.

- We generate synthetic data for a binary classification problem as follows: The feature vectors $\mathbf{z}_n$ are drawn independently from a multivariate Normal distribution $\mathbf{z}_n \sim \mathcal{N}(0, \boldsymbol{\Sigma})$ with covariances $\Sigma_{ij} = 1$ for $i \neq j$ and $\Sigma_{ii} = 2$. The labels $\mathrm{y}_n$ are generated as follows: $\mathrm{y}_n \mid \mathbf{z}_n \sim \text{Bernoulli}(p)$ if $\langle \mathbf{z}_n, 1 \rangle \leq 0$, otherwise $\mathrm{y}_n \mid \mathbf{z}_n \sim \text{Bernoulli}(1 - p)$. We generate $N = 10^4$ examples with $d = 2$ and $p = 0.1$.

- The CIFAR-10 dataset (Krizhevsky, 2009, Chapter 3) consists of $N = 60{,}000$ images of size $d = 3072$ in 10 classes. We consider a binary classification task: We set $\mathrm{y}_n = 1$ for the class "airplane", and $\mathrm{y}_n = -1$ for all others.

- The MNIST dataset (LeCun et al., 1998) consists of $N = 60{,}000$ images of handwritten digits of size $d = 784$. We consider a binary version of this task: We set $\mathrm{y}_n = 1$ for the digit 0, and $\mathrm{y}_n = -1$ for all others.

- The Iris dataset (Fisher, 1936) contains data of $N = 150$ examples of Iris flowers characterized by $d = 4$ numerical attributes. We consider the task of distinguishing Iris Setosa ($\mathrm{y}_n = 1$) from Iris Versicolour and Iris Virginica ($\mathrm{y}_n = -1$).

- The UCI ML Breast Cancer Wisconsin Diagnostic dataset (Dua & Graff, 2017), henceforth "Breast cancer", contains data for a binary classification task. It consists of $N = 569$ examples (357 negative, 212 positive) with $d = 30$ numerical features corresponding to various measurements of tumors. We set $\mathrm{y}_n = 1$ for malignant tumors, and $\mathrm{y}_n = -1$ for benign tumors.

- The KDD Cup '99 (Bay et al., 2000) dataset contains data for an intrusion detection task. It consists of 699,691 data with $d = 4$ attributes, of which 0.3% are anomalies ($\mathrm{y}_n = 1$). We sub-sample the dataset to $N = 70{,}000$.

**ScatterNet features**  In addition to the raw features, we also consider features extracted by a Scattering Network (ScatterNet) (Oyallon & Mallat, 2015). ScatterNet features have recently been demonstrated to be highly effective for differentially private image classification (Tramèr & Boneh, 2021). ScatterNet is an expansive, Wavelet-based transform that extracts features of size $(C \cdot 81, H/4, W/4)$ from images with $C$ channels, width $W$ and height $H$.

**Results**  First, we compare our adaptive schedules for DP-GD to constant schedules. We show the convergence of the gradient norm as a function of the privacy expenditure in Figure 1 for two datasets (rows) and two loss functions (columns). The plots are obtained by tracking the cumulative privacy cost across the iterations of the algorithm. The lines shown are the median of 120 repetitions. Recall that for the data-dependent schedule, the privacy cost is calculated on the assumption that the gradient norm is available at no privacy cost. Furthermore, we show the empirical risk at two privacy levels for all datasets in Table 3. The *Best constant* column shows the risk of the constant schedule with the best noise variance known in hindsight. The *Ours* column refers to the data-independent schedule.

|  | $\epsilon = 0.1$ | | $\epsilon = 20$ | |
|---|---|---|---|---|
|  | Ours | Best constant | Ours | Best constant |
| Synthetic | **0.5090** | 0.5307 | **0.5087** | 0.5087 |
| MNIST | 1.0058 | **0.9449** | 0.6510 | **0.5822** |
| Iris | **0.6465** | 0.6809 | **0.2778** | 0.2782 |
| Breast Cancer | 1.1656 | **0.8651** | **0.2399** | 0.2437 |
| KDD Cup '99 | **0.5864** | 0.5914 | **0.5401** | 0.5402 |

Table 3: Empirical risk for various datasets. *Best constant* refers to the constant schedule with best noise variance in hindsight. Lower value highlighted in bold.

Next, we compare to output perturbation on the CIFAR-10 dataset with ScatterNet features using $\ell_2$ regularization with $\lambda = 10$. The excess risk at various privacy levels is shown in Figure 2. The *excess* is with respect to the regularized non-private solution. *Best constant* again refers to the constant schedule with best noise variance in hindsight.

# 5 Discussion and Future Work

In this work, we have proposed the PUR as a criterion to select time-varying hyperparameters in differentially private iterative optimization algorithms. The PUR can be computed from a descent lemma, that is, a bound on the per-step objective improvement, and the per-step privacy loss associated with the selected hyperparameters. We have instantiated this framework for DP-GD on non-convex, convex and strongly convex functions, respectively. In this setting, the PUR-optimal hyperparameters achieve the same convergence rate as non-private GD in terms of iterations, and, in the case of strong convexity, also the optimal privacy-utility convergence. In the case of non-strongly convex functions, the privacy-utility convergence we have been able to establish is suboptimal. This might not be a limitation of PUR in general, but rather a consequence of choosing GD as the optimization algorithm. It is known that SGD has substantial privacy benefits over GD via privacy amplification by sub-sampling (Bassily et al., 2014) and by iteration (Feldman et al., 2020), and we expect that the PUR framework can be applied to these algorithms as well.

In general, PUR-optimal hyperparameters are data dependent, which makes the derivation of the privacy guarantee non-trivial. We have addressed this issue by substituting the gradient norm with an analytical upper bound, but other approaches are certainly conceivable. Note that the privacy leakage from the gradient norm is the highest when we are close to the optimum. This is because, when $\theta_t$ is sufficiently close to the optimum, there is a neighboring dataset on which $\theta_t$ is optimal. In that case, the norm of the neighboring gradient at $\theta_t$ is zero, hence, the added noise in the neighboring scenario would also be zero. A potential workaround to this problem could consist of enforcing a lower bound on the noise variance for small gradients, while letting the data-dependent term dominate for large gradients.

Due to the generality of the PUR framework, future work could apply it to a range of other optimization algorithms and objective families. Descent lemmas are known for a variety of optimization settings (Bertsekas, 1997; Bauschke et al., 2017; Korba et al., 2020; Khirirat et al., 2021; Arora et al., 2022). This suggests that PUR-optimal hyperparameters could be derived for non-smooth problems as well.

Finally, it might improve the hyperparameter selection to consider optimization problems over a longer time horizon $T > 1$. Ideally, we would like to minimize the excess loss under a privacy constraint. Although analytical bounds for excess loss are only available under strong assumptions (see e.g. Hong et al. (2022)), we may hope to find a tractable approximation numerically. A possible relaxation of this problem might be to minimize a weighted sum of privacy loss and utility improvement, as is common in the field of multi-objective optimization (Miettinen, 1998).

# 6 Related Work

A number of recent works have considered approaches to allocating the privacy budget non-uniformly across iterations in differentially private optimization. They broadly fall into two categories: (i) approaches that adapt the noise variance, and (ii) gradient-clipping approaches that adapt the clipping threshold. In this section, we summarize them and discuss their relation to our work.

**Adaptive noise** Lee & Kifer (2018) perform an adaptation of the noise variance and step size. The step size is chosen at each iteration by grid search over a predefined range via the Noisy Argmax mechanism. The noise variance is reduced by a constant factor whenever a noisy gradient does not lead to a decreased objective value. Yu et al. (2019) consider two strategies for adapting the noise variance, and compare them empirically for deep learning tasks. The strategies under investigation are (i) adjusting the noise variance periodically by monitoring the loss on a public validation dataset, and (ii) pre-defined schedules for the noise variance. The decay rate is found via differentially private model selection. A geometrically decaying noise variance has also been considered by Du et al. (2021), and by Zhang et al. (2021) for deep learning. Feldman et al. (2020) consider a variant of Proximal Noisy SGD with variable batch sizes, step sizes and noise variances. Their privacy guarantees make no strong assumptions on the noise sequence, but the convergence rate is derived for a constant noise sequence with varying batch size and step size. Their convergence guarantee holds for convex Lipschitz-continuous, Lipschitz-smooth objectives. Hong et al. (2022) derive a noise sequence by minimizing an analytical upper bound on the excess loss after $T$ steps. Their analysis requires a number

of assumptions on the loss function, namely convexity, Lipschitz-continuity, Lipschitz-smoothness and the Polyak-Lojasiewicz condition (Polyak, 1963). In contrast, our approach can be applied to any loss function for which a descent lemma can be established, which includes a much broader family of losses.

**Adaptive clipping**   Closely related to adaptive noise selection is the method of adaptive gradient clipping. Gradient clipping has been used in DP-ERM to make loss functions that do not have a bounded gradient amenable to gradient perturbation (Abadi et al., 2016). The privacy guarantee scales with the clipping threshold. Hence, adaptive gradient clipping is an alternative way to adjust the allocation of the privacy budget across iterations. For deep learning, Abadi et al. (2016) proposes to group the gradient components by the network layer they correspond to, and clip each group individually. Andrew et al. (2021) set the clipping threshold to a quantile of the gradient norm distribution. The quantile is estimated from past gradients via Online Gradient Descent (Shalev-Shwartz, 2012). While most works focus on the $\ell_2$ norm of the gradient, Pichapati et al. (2019) instead use a coordinate-wise adaptive clipping threshold. Song et al. (2020) study the convergence of adaptive clipping for (convex and non-convex) generalized linear models. Finally, we remark that there is an ongoing line of work studying the convergence properties of clipped SGD outside the context of differential privacy (Zhang et al., 2020b;a; Mai & Johansson, 2021).

Other adaptive differentially private optimizers include differentially private versions of AdaGrad (Duchi et al., 2010; 2011; Asi et al., 2021) and Adam (Kingma & Ba, 2015), which adapt to the problem geometry, and ADADP (Koskela & Honkela, 2020) which adapts the step size by thresholding the difference between a full gradient step and two half-steps.

### Acknowledgments

This work was partially supported by the Wallenberg AI, Autonomous Systems and Software Program (WASP) funded by the Knut and Alice Wallenberg Foundation. This research has been carried out as part of the Vinnova Competence Center for Trustworthy Edge Computing Systems and Applications at KTH Royal Institute of Technology. We would like to thank Tobias Oechtering and Joakim da Silva for helpful discussions.

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

## A    Additional figures

In Figure 3, we show the gradient norm plots for the experiment described in Section 4 for the remaining three datasets.

## B    Proofs

### B.1    Lemma 1

*Proof.* Since $F$ is $M$-smooth, we have

$$F(\theta_{t+1}) \leq F(\theta_t) - \eta_t \left(\nabla F(\theta_t) + \zeta_t\right)^\top \nabla F(\theta_t) + \frac{M}{2}\|\eta_t(\nabla F(\theta_t) + \zeta_t)\|^2$$

$$= F(\theta_t) - \eta_t \left(\|\nabla F(\theta_t)\|^2 + \zeta_t^\top \nabla F(\theta_t)\right) + \frac{M}{2}\|\eta_t(\nabla F(\theta_t) + \zeta_t)\|^2.$$

Note that the variance $\sigma_t^2$ of the noise $\zeta_t$ is itself a random variable. This is because we choose $\sigma_t$ as a function of $\theta_t$, which is random. Conditional on the values of $\theta_t$ and $\sigma_t$, the noise $\zeta_t$ has zero mean and variance $\sigma_t^2$, and is independent of $\nabla F(\theta_t)$. We take conditional expectation

$$\mathbb{E}\left[F(\theta_{t+1}) \mid \theta_t, \sigma_t\right] \leq \mathbb{E}\left[F(\theta_t) - \eta_t\left(\|\nabla F(\theta_t)\|^2 + \zeta_t^\top \nabla F(\theta_t)\right) + \frac{M}{2}\|\eta_t(\nabla F(\theta_t) + \zeta_t)\|^2 \mid \theta_t, \sigma_t\right]$$

$$= F(\theta_t) - \eta_t\|\nabla F(\theta_t)\|^2 + \frac{M}{2}\eta_t^2 \left(\|\nabla F(\theta_t)\|^2 + \mathbb{E}\left[\|\zeta_t\|^2 \mid \sigma_t\right]\right),$$

using $\mathbb{E}\left[\zeta_t^\top \nabla F(\theta_t) \mid \theta_t, \sigma_t\right] = \mathbb{E}\left[\zeta_t \mid \sigma_t\right]^\top \nabla F(\theta_t) = 0$. Now, we use $\mathbb{E}\left[\|\zeta_t\|^2 \mid \sigma_t\right] = d\sigma_t^2$ and rearrange to obtain:

$$\mathbb{E}\left[F(\theta_t) - F(\theta_{t+1}) \mid \theta_t, \sigma_t\right] \geq \left(\eta_t - \frac{M}{2}\eta_t^2\right)\|\nabla F(\theta_t)\|^2 - \frac{M}{2}\eta_t^2 d\sigma_t^2.$$

$\square$

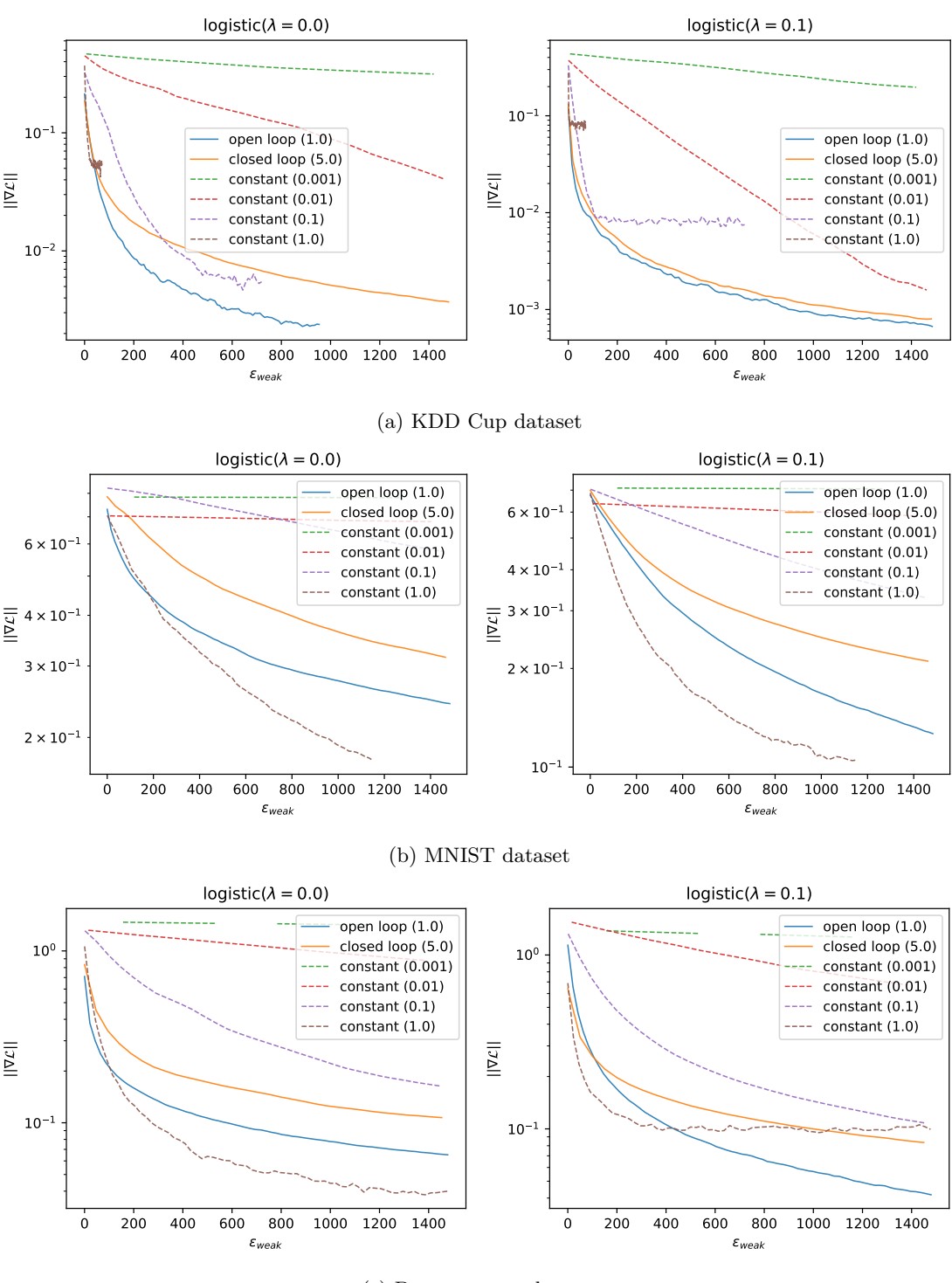

Figure 3: Results for additional datasets.

## B.2 Proposition 1

*Proof.* Since $P$ does not depend on $\eta_t$, we can first maximize $U$ with respect to $\eta_t$, which is attained by

$$\eta_t = \frac{1}{M\left(1 + d\sigma_t^2/\|\nabla F(\theta_t)\|^2\right)}. \tag{12}$$

Inserting this into Equation 3 leads to an expression of the form

$$\text{PUR}(\sigma_t) = c\left(\|\nabla F(\theta_t)\|^{-2}\sigma_t^{-1} + d\|\nabla F(\theta_t)\|^{-4}\sigma_t\right)$$

where $c$ is a constant. This is minimized by $\sigma_t = \|\nabla F(\theta_t)\|/\sqrt{d}$. Insertion into Equation 12 completes the result. $\qquad\square$

## B.3 Proposition 2

*Proof.* We start from Lemma 1 and insert the schedule from Equation 6 into Equation 5, which gives us a conditional expectation

$$\mathbb{E}\left[F(\theta_t) - F(\theta_{t+1}) \mid \theta_t, \sigma_t\right] \geq \frac{1}{4M}\|\nabla F(\theta_t)\|^2,$$

that is, the expected improvement taken over a single iteration. In order to average over the randomness of the entire algorithm, we apply the law of total expectation to obtain

$$\mathbb{E}\left[F(\theta_{t+1})\right] \leq \mathbb{E}\left[F(\theta_t)\right] - \frac{1}{4M}\mathbb{E}\left[\|\nabla F(\theta_t)\|^2\right]. \tag{13}$$

Now, we separate the three cases.

(a) We sum both sides of Equation 13 over $t = 0, \ldots, T$ to obtain

$$\mathbb{E}\left[F(\theta_{T+1})\right] \leq \mathbb{E}\left[F(\theta_0)\right] - \frac{1}{4M}\mathbb{E}\left[\sum_{t=0}^{T}\|\nabla F(\theta_t)\|^2\right].$$

We lower-bound $\mathbb{E}\left[F(\theta_{T+1})\right] \geq F^*$ and rearrange to obtain

$$\mathbb{E}\left[\frac{1}{T}\sum_{t=1}^{T}\|\nabla F(\theta_t)\|^2\right] \leq \frac{4M\left(F(\theta_0) - F^*\right)}{T}.$$

(b) By convexity, $F(\theta) - F(\theta^*) \leq \nabla F(\theta)^\top(\theta - \theta^*)$ for all $\theta \in \mathbb{R}^d$ and Cauchy-Schwartz implies that

$$F(\theta) - F(\theta^*) \leq \|\theta - \theta^*\|\,\|\nabla F(\theta)\|.$$

We take $\theta = \theta_t$ and invoke the assumption that $\|\theta_t - \theta^*\| \leq R$, which yields

$$\|\nabla F(\theta_t)\|^2 \geq \frac{(F(\theta_t) - F^*)^2}{R^2}. \tag{14}$$

Now, by taking expectation and applying Jensen's inequality, we have

$$\mathbb{E}\left[\|\nabla F(\theta_t)\|^2\right] \geq \frac{\mathbb{E}\left[F(\theta_t) - F^*\right]^2}{R^2}. \tag{15}$$

Plugging this bound back into Equation 13 yields the iterate relationship

$$\mathbb{E}\left[F(\theta_{t+1})\right] \leq \mathbb{E}\left[F(\theta_t)\right] - \frac{1}{4MR^2}\mathbb{E}\left[F(\theta_t) - F^*\right]^2.$$

Letting $V_t = \mathbb{E}\left[F(\theta_t) - F^*\right]$, we can write this as

$$V_{t+1} \leq V_t - \frac{1}{4MR^2}V_t^2$$

and apply Lemma 6 of Khirirat et al. (2021) to obtain

$$\frac{1}{\mathbb{E}\left[F(\theta_t) - F^*\right]} \geq \frac{1}{\mathbb{E}\left[F(\theta_0) - F^*\right]} + \frac{t}{4MR^2}$$

$$\frac{1}{\mathbb{E}\left[F(\theta_t) - F^*\right]} \geq \frac{t}{4MR^2}$$

$$\mathbb{E}\left[F(\theta_t) - F^*\right] \leq \frac{4MR^2}{t}.$$

(c) By strong convexity, $\|\nabla F(\theta_t)\|^2 \geq 2\mu\left(F(\theta_t) - F^*\right)$. Inserting this into Equation 13,

$$\mathbb{E}\left[F(\theta_{t+1})\right] \leq \mathbb{E}\left[F(\theta_t)\right] - \frac{\mu}{2M}\mathbb{E}\left[F(\theta_t) - F^*\right].$$

Letting $V_t = \mathbb{E}\left[F(\theta_t) - F^*\right]$, we can write this as

$$V_{t+1} \leq \left(1 - \frac{\mu}{2M}\right)V_t.$$

The result follows by recursion.

$\square$

### B.4   Lemma 2

**Lemma 2.** *Let $V_t$ be a sequence in $\mathbb{R}_{\geq 0}$ that satisfies*

$$V_{t+1} \leq V_t - qV_t^2 + \frac{r}{(t+1)^2}, \quad for \quad q > 0,\ 0 \leq r \leq \frac{1}{q},\ V_0 \leq \frac{1}{q}. \tag{16}$$

*Then,*

$$V_t \leq \frac{2}{qt}. \tag{17}$$

*Proof.* First, consider the upper bound in Equation 16 as a function $W_t : \mathbb{R} \to \mathbb{R}$ of $V_t$:

$$W_t(a) = -qa^2 + a + rt^{-2}.$$

$W_t$ is a concave quadratic maximized by

$$a^* = \arg\max_a W_t(a) = \frac{1}{2q}, \qquad W_t(a^*) = \frac{1}{4q} + rt^{-2}. \tag{18}$$

The proof is by induction. We begin by verifying that

$$V_1 \leq W_1(V_0) = \frac{1}{4q} + r \leq \frac{5}{4q} \leq \frac{2}{q}.$$

Now, assume that Equation 17 holds for some $t \geq 1$. We distinguish two cases.

First, if $t \geq 4$ then $V_t$ is smaller than the maximizer of $W_{t+1}$:

$$V_t \leq \frac{2}{qt} \leq \frac{1}{2q} = a^*.$$

Consequently, $W_{t+1}$ is monotonically increasing on $[0, \frac{2}{qt}]$, hence

$$V_{t+1} \leq W_{t+1}(V_t) \leq W_{t+1}\left(\frac{2}{qt}\right) = \frac{2}{qt} - \frac{4}{qt^2} + \frac{r}{(t+1)^2}.$$

Using the fact that $1/t \leq 1/t^2 + 1/(t+1)$, it follows that

$$\begin{aligned}
V_{t+1} &\leq \frac{2}{q}\left(\frac{1}{t^2} + \frac{1}{t+1}\right) - \frac{4}{qt^2} + \frac{r}{(t+1)^2} \\
&\leq \frac{2}{q(t+1)} - \frac{1}{qt^2} \\
&\leq \frac{2}{q(t+1)},
\end{aligned}$$

where the second inequality follows from $r \leq 1/q$.

Second, if $t \leq 3$ then we can use the global maximizer to bound $V_{t+1}$:

$$V_{t+1} \leq W(a^*) = \frac{1}{4q} + r(t+1)^{-2} = \frac{(t+1)^2 + 4rq}{4q(t+1)^2} = \frac{2}{q(t+1)} + \frac{(t+1)^2 - 8(t+1) + 4rq}{4q(t+1)^2}.$$

The numerator of the second term is a convex quadratic. Over $t \in [0,3]$, it is maximized by $t = 0$, which leads to

$$V_{t+1} \leq \frac{2}{q(t+1)} + \frac{4rq - 7}{4q(t+1)^2}.$$

We can see that second term is negative because $rq \leq 1$, so it can be dropped to conclude $V_{t+1} \leq 2/q(t+1)$.
$\square$

### B.5 Proposition 3

*Proof.* As with Proposition 2, the general proof idea is to apply the descent lemma to the schedule and then bound the various quantities in order to arrive at a recursive bound on $\mathbb{E}\left[F(\theta_t) - F^*\right]$. Again, we separate the convex and strongly convex case.

(a) Starting from Lemma 1, inserting the schedule $\sigma_t = \frac{4M\sqrt{F(\theta_0) - F^*}}{\sqrt{dt}}$ and taking expectation leads to

$$\mathbb{E}\left[F(\theta_{t+1})\right] \leq \mathbb{E}\left[F(\theta_t)\right] - \frac{3}{8M}\mathbb{E}\left[\|\nabla F(\theta_t)\|^2\right] + \frac{2M(F(\theta_0) - F^*)}{t^2}.$$

Now, we sum both sides over $t$ to obtain

$$\mathbb{E}\left[F(\theta_{T+1})\right] \leq \mathbb{E}\left[F(\theta_0)\right] - \frac{3}{8M}\mathbb{E}\left[\sum_{t=1}^T \|\nabla F(\theta_t)\|^2\right] + \sum_{t=1}^T \frac{2M(F(\theta_0) - F^*)}{t^2}.$$

Noting that $\sum_{t=1}^T 1/t^2 \leq \sum_{t=1}^\infty 1/t^2 = \pi^2/6$, we bound the last term to obtain

$$\mathbb{E}\left[F(\theta_{T+1})\right] \leq \mathbb{E}\left[F(\theta_0)\right] - \frac{3}{8M}\mathbb{E}\left[\sum_{t=1}^T \|\nabla F(\theta_t)\|^2\right] + \frac{\pi^2 M(F(\theta_0) - F^*)}{3}.$$

Now, we lower-bound $F(\theta_{T+1}) \geq F^*$ and rearrange to obtain

$$\mathbb{E}\left[\frac{1}{T}\sum_{t=0}^T \|\nabla F(\theta_t)\|^2\right] \leq \frac{8(\pi^2 M/3 + 1)(F(\theta_0) - F^*)}{3T}.$$

(b) Starting from Lemma 1, inserting the schedule $\sigma_t = \frac{4MR}{\sqrt{dt}}$ and $\eta_t = 1/(2M)$ and taking expectation leads to

$$\mathbb{E}\left[F(\theta_{t+1})\right] \leq \mathbb{E}\left[F(\theta_t)\right] - \frac{3}{8M}\mathbb{E}\left[\|\nabla F(\theta_t)\|^2\right] + \frac{2MR^2}{t^2}. \tag{19}$$

Analogously to Equation 15, we can bound $\mathbb{E}\left[\|\nabla F(\theta_t)\|^2\right]$ to obtain

$$\mathbb{E}\left[F(\theta_{t+1})\right] \leq \mathbb{E}\left[F(\theta_t)\right] - \frac{3}{8MR^2}\mathbb{E}\left[F(\theta_t) - F^*\right]^2 + \frac{2MR^2}{t^2},$$

which we can write as

$$V_{t+1} \leq V_t - \frac{3}{8MR^2}V_t^2 + \frac{2MR^2}{t^2}.$$

Applying Lemma 2 yields the result.

(c) We apply Lemma 1 with the schedule $\sigma_t = \sqrt{2\mu\left(F(\theta_0) - F^*\right)\left(1 - \mu/(2M)\right)^t/d}$ and $\eta_t = 1/(2M)$, and take expectation to obtain

$$\mathbb{E}\left[F(\theta_{t+1})\right] \leq \mathbb{E}\left[F(\theta_t)\right] - \frac{3}{8M}\mathbb{E}\left[\|\nabla F(\theta_t)\|^2\right] + \frac{1}{8M}(2\mu)(F(\theta_0) - F^*)\left(1 - \frac{1}{2\kappa}\right)^t,$$

where we write $\kappa = M/\mu$. We bound the gradient norm by $\|\nabla F(\theta_t)\|^2 \geq 2\mu\left(F(\theta_t) - F^*\right)$ due to strong convexity:

$$\mathbb{E}\left[F(\theta_{t+1})\right] \leq \mathbb{E}\left[F(\theta_t)\right] - \frac{3}{4\kappa}\mathbb{E}\left[F(\theta_t) - F^*\right] + \frac{1}{4\kappa}(F(\theta_0) - F^*)\left(1 - \frac{1}{2\kappa}\right)^t. \tag{20}$$

Now we can show the result by induction. Suppose it is true for some $t \geq 1$ that $\mathbb{E}\left[F(\theta_t) - F^*\right] \leq (F(\theta_0) - F^*)\left(1 - \frac{1}{2\kappa}\right)^t$. Then, we can subtract $F^*$ from both sides of Equation 20 and apply the induction hypothesis to obtain

$$\mathbb{E}\left[F(\theta_{t+1}) - F^*\right] \leq \left(1 - \frac{3}{4\kappa} + \frac{1}{4\kappa}\right)(F(\theta_0) - F^*)\left(1 - \frac{1}{2\kappa}\right)^t$$
$$= (F(\theta_0) - F^*)\left(1 - \frac{1}{2\kappa}\right)^{t+1},$$

which is what we wanted to show. The initial case $t = 0$ can be verified via the descent lemma. $\qquad\square$

### B.6 Proposition 4

*Proof.* We begin with the privacy analysis. We first analyze the privacy loss in terms of zCDP, then convert to the corresponding $(\epsilon, \delta)$-DP. Each iteration of the algorithm is an application of the Gaussian mechanism to the gradient $\nabla F(\theta_t) = 1/N \sum_n f(\theta_t; \boldsymbol{x}_n)$. Since $f$ is $L$-Lipschitz, the sensitivity of $\nabla F$ with respect to replacement of one data entry is $\Delta = 2L/N$. Adding noise with variance $\sigma_t^2$ preserves $\rho_t$-zCDP with $\rho_t = \Delta^2/(2\sigma_t^2) = 2(L/N\sigma_t)^2$. Composition over $T$ iterations means that the full algorithm preserves $\rho$-zCDP with

$$\rho = \frac{2L^2}{N^2}\sum_{t=1}^{T}\frac{1}{\sigma_t^2}. \tag{21}$$

Now we specialize this guarantee for the noise schedules under consideration.

For (a) we have

$$\sum_{t=1}^{T}\frac{1}{\sigma_t^2} = \frac{d}{16M^2(F(\theta_0) - F^*)}\sum_{t=1}^{T}t^2.$$

Note that $\sum_t t^2 \leq T^3$. Plugging back into Equation 21, we have

$$\rho \leq \frac{L^2 d}{8N^2 M^2 (F(\theta_0) - F^*)} T^3.$$

That is, we can run $T = \sqrt[3]{8\rho N^2 M^2 (F(\theta_0) - F^*)/(L^2 d)}$ iterations until the privacy budget is exhausted. Inserting this into the excess risk bound from Proposition 3, we conclude

$$\mathbb{E}\left[\frac{1}{T}\sum_{t=0}^{T}\|\nabla F(\theta_t)\|^2\right] \leq \frac{8(\pi^2 M/3 + 1)(F(\theta_0) - F^*)}{3\sqrt[3]{8\rho N^2 M^2 (F(\theta_0) - F^*)/(L^2 d)}}$$

$$\leq \frac{8}{3}\left(\frac{\pi^2 M}{3} + 1\right)\sqrt[3]{\frac{(F(\theta_0) - F^*)^2 L^2 d}{8\rho N^2 M^2}}.$$

For (b), the proof is analogous to (a), but using the noise schedule and excess risk bound for the convex case.

For (c) the argument follows a similar structure. The algorithm is $\rho$-zCDP for

$$\rho = \frac{d\,\Delta^2/2}{2\mu\,(F(\theta_0) - F^*)}\sum_t\left(1 - \frac{1}{2\kappa}\right)^{-t}$$

where $\kappa = M/\mu$. This is a geometric series $\rho = a\sum_{t=1}^{T} r^t$ with constants $a = \frac{d\Delta^2/2}{2\mu F(\theta_0) - F^*}$ and $r = 1/(1 - 1/(2\kappa))$. Therefore,

$$\rho = a\frac{r^{T+1} - r}{r - 1}$$

$$= a\frac{r}{r - 1}(r^T - 1).$$

Note that $r/(r - 1) = 2\kappa$, therefore

$$\rho = 2a\kappa(r^T - 1)$$

$$\leq 2a\kappa r^T$$

$$= \frac{d\Delta^2}{2\mu\,(F(\theta_0) - F^*)}\left(1 - \frac{1}{2\kappa}\right)^{-T}.$$

Via Proposition 3b, we have

$$\rho \leq \frac{d\Delta^2}{2\mu\mathbb{E}\left[F(\theta_T) - F^*\right]}$$

$$\mathbb{E}\left[F(\theta_T) - F^*\right] \leq \frac{d\Delta^2}{2\mu\rho}.$$

The corresponding $(\epsilon, \delta)$-DP guarantees are $\epsilon = \rho + 2\sqrt{\rho\log\delta^{-1}}$.

$\square$

