# OpenReview forum: "Adaptive Hyperparameter Selection for Differentially Private Gradient Descent"
_TMLR — Accepted by TMLR_

### Review · Reviewer_g8Jn · 2023-05-09

**Summary Of Contributions:**

The paper makes several contributions:
1. A conceptual framework for tuning hyperparameters by optimizing the privacy-utility ratio.
2. Use this framework on convex and strongly convex losses.
3. Give convergence rate for both data-independent and data-dependent scheduling.
4. Provide extensive experiments to substantiate their results.



**Audience:**

Yes

**Claims And Evidence:**

Yes

**Requested Changes:**

The paper is well written and I approve of accepting the paper in the current format.

**Strengths And Weaknesses:**

The paper studies an important problem under the most commonly studied assumptions. The two hyperparameters considered in this paper are the learning rate and the step size.

The paper is very well written, and quiet easy to ready and follow. I really liked this aspect of the paper. The proofs all seem to be correct.

Can the authors give a reason why they have a worse excess risk guarantee for convex, Lipschitz, and Lipschitz smooth functions?

---

> ### Author Response · Authors · 2023-06-08
> **Response to Reviewer g8Jn**
>
> We would like to express our gratitude to the Reviewer for their thoughtful review, their positive feedback and insightful comments.
>
> It is correct that, in the convex case, the excess risk bound achieved by the PUR-optimal schedules is somewhat higher than bounds found in prior work. Our bound scales with the cubic root instead of the square root. In essence, this is because of the fast convergence rate of the PUR-optimal schedules. The privacy loss grows at a rate of $O(t^3)$ while the excess loss only converges at a rate of $O(1/t)$. Since $O(1/t)$ is also the convergence rate of non-private gradient descent, this tells us that the only way to improve the excess risk bound would be to slow down the decrease of $\sigma_t$. For instance, if we picked $\sigma_t = \Theta(1/\sqrt{t})$ then the privacy loss would grow at a rate of $O(t^2)$. However, we can see that a rate of $\sigma_t = \Theta(1/\sqrt{t})$ is not PUR-optimal, because, according to Proposition 1, $\sigma_t$ should be proportional to the gradient norm. We can see that this is not satisfied, because Proposition 2(a) (in the revised version) tells us that the sum of squared gradient norms converges as $T \to \infty$, whereas $\sum_t^T \sigma_t^2 = \sum_t^T 1/t$ does not converge.

---

### Review · Reviewer_WeLj · 2023-05-14

**Summary Of Contributions:**

This work proposes an adaptive mechanism to select two hyperparameters in DP optimization: the noise scale and learning rate. In particular, the authors introduce PUR to maximize the trade-off between utility and privacy. The proposed method is supported in theory and experiments under the convex (even sometimes strongly convex) losses.

**Audience:**

Yes

**Claims And Evidence:**

Yes

**Requested Changes:**

See weaknesses above.

**Strengths And Weaknesses:**

Strengths: This work is clearly written and rigorous as far as I can tell, e.g. the discussion of constraint $\epsilon<1$ in Section 3.2 and the careful discussion from the data-dependent selection to the data-independent one. Especially, the introduction of privacy-utility ratio (PUR) in Section 3 is interesting. I am impressed by the elegant selection of $\sigma$ and $\eta$ in Equation (6) and the similarity in performance between data-dependent and data-independent selection.

Weaknesses: One limitation of this work is the convex (and smooth) setting. While I understand that this is necessary to derive theoretical results, the experiments are much less significant since the model is a simple logistic regression and does not reflect the necessity of a better hyperparameter selection. In other words, there are several aspects missing or good-to-have in this work. (1) In the convex loss setting, one can still consider the neural networks, which is the motivation in the first sentence of the first section. I.e., the hyperparameter selection is more of an issue when the model is larger. Notice that in linear probing (fine-tuning only the last layer of a network), this is essentially a convex setting and can lead to state-of-the-art accuracy in computer vision (e.g. https://arxiv.org/abs/2205.02973). This would enhance the paper to go beyond toy datasets like MNIST. (2) Beyond the convex loss setting, e.g. the deep learning setting, it seems possible to derive a result similar to (https://arxiv.org/abs/2206.07136) following the analysis in this work. (3) There are non-gradient based methods to solve a logistic regression in a differentially private way, to which this work should be compared.

---

> ### Author Response · Authors · 2023-06-08
> **Response to Reviewer WeLj**
>
> We would like to express our gratitude to the Reviewer for their thoughtful review and constructive feedback.
>
> Regarding the comment that our results are limited to convex problems. We have conducted further analysis and are pleased to say that we have achieved similar convergence results for non-convex problems as well. We have incorporated these findings in Propositions 2, 3 and 4 in the revised version of the paper. We would like to express our gratitude for this insightful comment, as it has been instrumental in enhancing the quality and scope of our research.
>
> We appreciate the Reviewer's perspective on the importance of conducting additional experiments. We have decided to include two new experiments in the paper that we believe will contribute to its overall relevance. Firstly, as suggested by the Reviewer, we will include experiments involving neural networks, which will provide valuable insights and broaden the scope of our research. Additionally, we recognize that it would be valuable to provide a comparison to methods that are not based on gradient perturbation. Therefore, we will also compare our results with output perturbation (which was also suggested by Reviewer 455j). Due to the short discussion period, we have not managed to finalize these experiments yet, but we are working on them and will include them in the final version.
>
> We sincerely thank the Reviewer for their valuable input, as it has influenced our decision to incorporate these significant additions to the paper.

---

### Review · Reviewer_455j · 2023-05-26

**Summary Of Contributions:**

This paper studies the problem of differentially private gradient descent (DP-GD) in (strongly) convex settings. More specifically, the authors propose using an adaptive way to add random noise in DP-GD, with the goal of attaining improved utility guarantees in the theoretical analysis as well as practical applications.

**Audience:**

Yes

**Claims And Evidence:**

Yes

**Requested Changes:**

I have the following main questions that need to be addressed in the current paper:
1. More justification needs for the privacy-utility ratio framework. The current form lacks consideration of the iteration number, which accounts for the main part of the privacy loss.
2. According to Proposition 4, the results hold for specific $\epsilon$ since $\epsilon$ further depends on other parameters (e.g., d, L, T, etc.). Can you give a result that holds for any choice of $\epsilon$?
3. According to Proposition 4, the privacy guarantee relies on strong assumptions. For example, it requires the function to be smooth, which is not required by the previous methods. If we can assume that the function is smooth, we can directly use the output perturbation methods in [1] to achieve strong privacy and utility guarantees for DP-GD. In this case, we don't even need to tune anything. In addition, it is very hard to implement the proposed method since we don't know the smoothness parameter for general problems.
4. According to Proposition 4, how can the condition $\\|\theta_t-\theta^*\\|\leq R$ hold for convex problems?
5. For the experiments, the authors should consider comparing with the methods developed in [1] and Lee & Kifer (2018).
6. In general, I don't understand the benefits of the proposed method. On the one hand, the results do not improve over previous guarantees. On the other hand, we still need to tune hyper-parameters in practice since the smoothness parameter is unknown for general problems.

References:
1. Zhang, Jiaqi, et al. "Efficient private ERM for smooth objectives." Proceedings of the 26th International Joint Conference on Artificial Intelligence. 2017.

**Strengths And Weaknesses:**

Strengths of the paper:
1. The proposed privacy-utility ratio framework seems interesting and can be an independent interest.
2. The privacy and utility guarantees of the proposed adaptive approach have been established.
3. Experiments validate the effectiveness of the proposed method.

Weakness of the paper:
1. There is no improvement in the theoretical utility guarantees compared to existing methods.
2. The privacy guarantees rely on the smoothness assumption.
3. There is no comparison with some existing baselines.

---

> ### Author Response · Authors · 2023-06-08
> **General response to Reviewer 455j**
>
> We would like to thank the Reviewer for her/his insightful comments and suggestions as well as the careful review that has helped us modify the paper. We submit a separate comment for each of the reviewer's questions.

---

> ### Author Response · Authors · 2023-06-08
> **Question 1 of Reviewer 455j**
>
> > 1. More justification needs for the privacy-utility ratio framework. The current form lacks consideration of the iteration number, which accounts for the main part of the privacy loss.
>
> Thank you for this suggestion. We agree that the privacy-utility ratio could have been motivated a little more carefully, but please note that the noise allocation depends on both the iteration index (explicitly in the data-independent case, and implicitly in the data-dependent case) and the total number of iterations (through the total privacy budget constraint). Let us elaborate.
>
> Essentially, our problem is one of optimally allocating noise variances over multiple steps of a (stochastic) gradient process. The theory of dynamic programming tells us that the optimal solution to such a problem should, in general, not only account for the current stage cost, but also the cost to go from the resulting state. In this respect, our approach is greedy, as it only accounts for the expected progress in the present iteration. It is possible that the optimal solution to the multi-period allocation problem would have a more explicit dependence on iteration index than our current solution. However, it is not trivial to solve the multi-period allocation in an optimal way, and we would like to defer it to future work.
>
> The privacy allocation problem is of a multi-objective nature: increasing the noise variance improves the privacy, but impairs the progress that the optimization process can be expected to make. We have decided to optimize the ratio between privacy and utility, but one could also try to maximize the (weighed) difference between the expected progress and the privacy cost. If we compare with modern portfolio theory, such a weighted cost would correspond to the Markowitz objective (expected return - weighted risk), while our approach would correspond to maximizing the Sharpe ratio. Both approaches have found their use in portfolio theory, and it is possible that both approaches could be used to devise differentially private learning algorithms.

---

> ### Author Response · Authors · 2023-06-08
> **Question 2 of Reviewer 455j**
>
> > 2. According to Proposition 4, the results hold for specific $\epsilon$ since $\epsilon$ further depends on other parameters (e.g., d, L, T, etc.). Can you give a result that holds for any choice of $\epsilon$?
>
> The time horizon $T$ can be chosen such as to satisfy any desired level of privacy. We have added the number of iterations to Proposition 4 to clarify this.

---

> ### Author Response · Authors · 2023-06-08
> **Question 3 of Reviewer 455j**
>
> > 3. According to Proposition 4, the privacy guarantee relies on strong assumptions. For example, it requires the function to be smooth, which is not required by the previous methods. If we can assume that the function is smooth, we can directly use the output perturbation methods in [1] to achieve strong privacy and utility guarantees for DP-GD. In this case, we don't even need to tune anything. In addition, it is very hard to implement the proposed method since we don't know the smoothness parameter for general problems.
>
> It is true that the set-up where a single node trains a model to subsequently release it in a differentially private way can be addressed in many ways. Not only using gradient perturbation, as we do in our paper, but one could also imagine using objective or model perturbation. As pointed out by the reviewer, the privacy and utility of such output perturbation [1] would indeed result in similar results as ours.  We find it quite remarkable that it is possible to protect every computed gradient and still attain the same utility as output perturbation. In contrast to output perturbation, our approach can separate gradient evaluations (e.g. performed by the owner of the data) from the model training (performed by another party) and could possibly be extended to federated learning settings with multiple clients and local differential privacy.

---

> ### Author Response · Authors · 2023-06-08
> **Question 4 of Reviewer 455j**
>
> > 4. According to Proposition 4, how can the condition $\Vert \theta_t - \theta^* \Vert \leq R$ hold for convex problems?
>
> It is indeed a good observation that the boundedness of the iterates is not easy to guarantee in general, thank you for pointing this out. We have included an additional convergence result to Propositions 2, 3 and 4 that does not rely on bounded iterates or convexity. The result is stated in terms of the gradient norm. In the convex case, the assumption allows us to relate the gradient norm to the excess loss. A common way to enforce the assumption is via projection when the problem has a norm constraint. Proximal gradient methods can be used in this scenario and they often have similar convergence properties.

---

> ### Author Response · Authors · 2023-06-08
> **Question 5 of Reviewer 455j**
>
> > 5. For the experiments, the authors should consider comparing with the methods developed in [1] and Lee & Kifer (2018).
>
> We agree that output perturbation and Lee & Kifer (2018) are suitable methods to compare against and would improve our paper. We will include a comparison in the final version of the paper.

---

> ### Author Response · Authors · 2023-06-08
> **Question 6 of Reviewer 455j**
>
> > 6. In general, I don't understand the benefits of the proposed method. On the one hand, the results do not improve over previous guarantees. On the other hand, we still need to tune hyper-parameters in practice since the smoothness parameter is unknown for general problems.
>
> We hope that the extended theoretical results which we provide in the now revised version of the paper have helped to clarify the relevance of our proposed method. As described in our comment to Question 3, we believe that the generality of gradient perturbation makes it an interesting method even though similar results can be obtained with output perturbation in certain settings.

---

### Author Response · Authors · 2023-06-08
**Revised paper and comments**

We would like to thank the editor for handling our submission, and the reviewers for their helpful and constructive comments which have helped improve several aspects of the paper:

- On the technical side, the reviewers have made many helpful and constructive suggestions. We have made a revision that addresses several of these points. In particular, we have
  - followed the suggestion to provide results for more general classes of loss functions. We have extended Propositions 2, 3 and 4 with convergence results that do not rely on convexity or boundedness of iterates; and
  - re-formulated Proposition 4 to allow the time horizon T to be chosen as a function of any arbitrary desired privacy level.
- The reviewers have also made good suggestions for additional experiments. We are currently preparing an experiment on last-layer fine-tuning, as well as a comparison to output perturbation and Lee & Kifer (2018). We will include the results in the final version of the paper.

---

### Decision · Action_Editors · 2023-07-08

**Recommendation:** Accept with minor revision

**Comment:**

All reviewers recommend accepting the paper, on the condition that the authors add the experiments they have promised to add in the final version.

I thus recommend acceptance conditioned on minor revision of adding the promised new experiments.

**Audience:**

The reviewers believe the paper is of interest to the DP community.

**Claims And Evidence:**

All reviewers agree that the claims made in the submission are supported by sufficient evidence, subject to authors adding the experiments they have promised to add in the final version.